# Acidic pH reduces agonist efficacy and responses to synaptic-like glycine applications in zebrafish $\alpha$1 and rat $\alpha$1$\beta$ recombinant glycine receptors

Josip Ivica[1] 🆔, Remigijus Lape[2] 🆔 and Lucia G. Sivilotti[1] 🆔

[1]*Department of Neuroscience, Physiology and Pharmacology, University College London, London, UK*
[2]*Neurobiology Department, MRC Laboratory of Molecular Biology Cambridge Biomedical Campus Francis Crick Avenue, Cambridge, UK*

Edited by: David Wyllie & Jian Shi

The peer review history is available in the Supporting Information section of this article (https://doi.org/10.1113/JP282171#support-information-section).

**Abstract**   Many pentameric ligand-gated ion channels are modulated by extracellular pH. Glycine receptors (GlyRs) share this property, but it is not well understood how they are affected by pH changes. Whole cell experiments on HEK293 cells expressing zebrafish homomeric $\alpha$1 GlyR confirmed previous reports that acidic pH (6.4) reduces GlyR sensitivity to glycine, whereas alkaline pH (8.4) has small or negligible effects. In addition to that, at pH 6.4 we observed a reduction in the maximum responses to the partial agonists $\beta$-alanine and taurine relative to the full agonist glycine. In cell-attached single-channel recording, low pH reduced agonist efficacy, as the maximum open probability decreased from 0.97, 0.91 and 0.66 to 0.93, 0.57 and 0.34 for glycine, $\beta$-alanine and taurine, respectively, reflecting a threefold decrease in efficacy equilibrium constants for all three agonists. We also tested the effect of pH 6.4 in conditions that replicate those at the native synapse, recording outside-out currents elicited by fast application of millisecond pulses of agonists on $\alpha$1 and $\alpha$1$\beta$ GlyR, at a range of intracellular chloride concentrations. Acidic pH reduced the area under the curve of the currents, by reducing peak amplitude, slowing activation and speeding deactivation.

**Josip Ivica** is a physicist by training and completed his PhD in the Electronics and Computer Sciences Department of Southampton University, where he used planar lipid bilayers with $\alpha$-haemolysing nanopores to detect cancer-related microRNA. Currently, he works as a postdoctoral research associate in Lucia Sivilotti's lab at University College London. He is interested in the relation between structure and function in ligand gated ion channels, with a special focus on the functional characterisation of glycine receptors with electrophysiological techniques, including single-channel recording.

The Journal of Physiology

Our results show that acidification of the extracellular pH by one unit, as may occur in pathological conditions such as ischaemia, impairs GlyR gating and is likely to reduce the effectiveness of glycinergic synaptic inhibition.

(Resubmitted 19 October 2021; accepted after revision 16 November 2021; first published online 21 November 2021)

**Corresponding author** Lucia G. Sivilotti: Department of Neuroscience, Physiology and Pharmacology, University College London, London, WC1E 6BT, UK. Email: l.sivilotti@ucl.ac.uk

**Abstract figure legend** The function of recombinant glycine receptors is inhibited by a moderate acidic shift in extracellular pH, such as occurs after ischaemia or seizures. Left panel: at pH 6.4 (red), the agonist sensitivity of $\alpha 1$ GlyRs decreases: dose-response curves are shifted to higher concentrations and reach a lower relative maximum for the partial agonist taurine. Middle panel: the maximum open probability of clusters of single-channel $\alpha 1$ GlyR openings elicited by high agonist concentrations in cell-attached patches is lower in acidic pH (red), confirming a reduction in agonist efficacy. Right panel: acidic pH is likely to reduce glycinergic synaptic signals, as currents evoked by fast piezo stepper applications (2 ms, 3 mM glycine) to outside-out patches containing recombinant $\alpha 1 \beta$ GlyR reach a smaller peak and decay more quickly at pH 6.4.

**Key points**

- Extracellular pH in the central nervous system (CNS) is known to shift towards acidic values during pathophysiological conditions such as ischaemia and seizures.
- Acidic extracellular pH is known to affect GABAergic inhibitory synapses, but its effect on signals mediated by glycine receptors (GlyR) is not well characterised.
- Moderate acidic conditions (pH 6.4) reduce the maximum single channel open probability of recombinant homomeric GlyRs produced by the neurotransmitter glycine or other agonists, such as $\beta$-alanine and taurine.
- When glycine was applied with a piezoelectric stepper to outside out patches, to simulate its fast rise and short duration at the synapse, responses became shorter and smaller at pH 6.4. The effect was also observed with physiologically low intracellular chloride and in mammalian heteromeric GlyRs.
- This suggests that acidic pH is likely to reduce the strength of inhibitory signalling at glycinergic synapses.

## Introduction

Extracellular pH in the CNS is a dynamic value that can drift in either direction from its physiological value of 7.3, even during normal neuronal activity (Chesler, 1990, 2003). Common causes of these changes include release of protons from the content of synaptic vesicles and bicarbonate efflux upon activation of GABA and glycine ion channels (GlyRs) (Kaila & Voipio, 1987; Luckermann *et al.* 1997). More substantial changes occur in pathological conditions such as ischaemia or epileptiform activity, when pH can drop by more than one unit (Somjen, 1984; Siesjo *et al.* 1985; Katsura *et al.* 1991; Nedergaard *et al.* 1991; Hoffman *et al.* 1999). Acidosis adds its own effects to those of the primary insult to neurones and may contribute to excitotoxic neuronal death (Choi, 2020). An acidic environment modulates the activity of a wide range of voltage- and ligand-gated ion channels (Obara *et al.* 2008; Sinning & Hubner, 2013).

pH changes have diverse effects on $GABA_A$ receptors, which mediate most of the fast synaptic inhibition in the brain. Depending on the receptor subunit composition, acidic extracellular pH can potentiate, inhibit, or have no effect on GABA responses (Krishek *et al.* 1996). These differences result in complex effects on phasic and tonic GABA inhibitory signalling in different areas of the brain.

The picture is much simpler for the effects of pH on GlyR, which mediate much of fast inhibition in brainstem and spinal cord (Lynch, 2004; Bowery & Smart, 2006). Glycinergic responses have been reported to be inhibited at acidic pH and modestly enhanced by alkaline pH (Harvey *et al.* 1999; Li *et al.* 2003; Chen *et al.* 2004; Chen & Huang, 2007). There are no differences across GlyR receptor subtypes, possibly because of the strong homology across the relatively few GlyR subunit isoforms and the conservation across $\alpha$ subunits of the residues responsible for the effects of pH.

While the direction of pH modulation of GlyR seems well established, the picture lacks detail. Most of what we know of the effects of pH comes from experiments with slow agonist applications, where responses are distorted by desensitisation. This type of data does not allow us to estimate pH effects (if any) on the maximum open probability of the channel and is also a poor predictor of pH effects on synaptic currents, given that in the cleft GlyRs are exposed to high glycine concentrations (~3 mM), which rise quickly and last less than 1 ms (Beato, 2008). Conversely, experiments that measure native glycinergic synaptic currents in brain slices have the problem of poor pH control and small signals, especially if a physiological low chloride pipette internal solution is used.

Here we report the effect of one-unit changes in extracellular pH on the GlyR maximum open probability elicited by the full agonist glycine and by a set of partial agonists. We also characterised the effects of pH changes on the kinetics of current responses elicited by fast application of high agonist concentrations, a protocol designed to replicate synaptic conditions, at a range of internal chloride concentrations and including both a structural model for GlyR (zebrafish homomeric $\alpha 1$) and a synaptic-type GlyR (heteromeric rat GlyR). Our results clearly show that at acidic pH glycinergic currents are smaller, because of a smaller peak, a slower activation and a faster decay and that these effects are associated with an impairment in channel gating.

## Methods

### Cell culture and transfection

HEK293 cells (ATCC) were maintained in a humidified incubator at 95% air/5% $CO_2$ at 37°C in Dulbecco's modified Eagle's medium (DMEM) supplemented with 10% (vol/vol) heat-inactivated fetal bovine serum, 100 U/ml penicillin and 100 U/ml streptomycin sulphate (all from GIBCO, UK). Cells were passaged after reaching 70–85% confluence, on average every 2–3 days and up to 25 times. For transfection, HEK293 cells were plated on 13 mm poly-L-lysine-coated coverslips in 35 mm Petri dishes and transfected by the $Ca^{2+}$ phosphate DNA coprecipitation method (Groot-Kormelink *et al.* 2002) using a total of 3 $\mu$g of DNA for each dish. In order to express the zebrafish homomeric $\alpha 1$ GlyR, the mixture consisted of pcDNA3 plasmids with inserts coding for enhanced green fluorescence protein (eGFP) and for WT zebrafish $\alpha 1$ GlyR (UniProt accession number O93430) and the final composition was 5% WT zebrafish GlyR, 20% eGFP and 75% of empty pcDNA3 plasmid. The empty plasmid was added to optimise the expression level (Groot-Kormelink *et al.* 2002). For experiments with the rat heteromeric $\alpha 1 \beta$ receptor, cells were transfected with plasmids coding for the rat $\alpha 1$ (GenBank accession number AJ310834) and $\beta$ subunits (GenBank accession number AJ310839) at a ratio of 1:40. The final mixture of cDNA plasmids contained 1% $\alpha 1$, 40% $\beta$, 20% eGFP and 39% of empty pcDNA3 plasmid.

Cells were washed 8 h after transfection and electrophysiological experiments were performed 24–48 h after transfection.

### Electrophysiological experiments

Electrodes for electrophysiological recording were made from thick-walled borosilicate capillaries (GC150F-7.5; Harvard Apparatus, UK) with a Sutter P-97 pipette puller (Sutter Instruments Co., USA). Subsequently, pipette tips were fire-polished at a microforge to obtain a final resistance of 3–5 MΩ (whole-cell recordings) and 5–12 MΩ (cell-attached and outside-out recordings).

The bath extracellular solution contained (in mM): 112.7 NaCl, 20 sodium gluconate, 2 KCl, 2 $CaCl_2$, 1.2 $MgCl_2$, 10 tetraethylammonium Cl (TEACl), 30 glucose and 10 HEPES; the pH was adjusted with NaOH to 7.4. For electrophysiological recordings with a different pH value, 10 mM HEPES was replaced with 10 mM 2-(*N*-morpholino)ethanesulfonic acid (MES) or 10 mM *N*-Tris(hydroxymethyl)methyl-4-aminobutanesulfonic acid (TABS) and adjusted with NaOH to pH 6.4 or pH 8.4, respectively. This was done to increase buffering capacity since the buffers perform the best when the pH is close to their p$K_a$ value. HEPES, MES and TABS have p$K_a$ values (at 25°C) of 7.48, 6.1 and 8.9 respectively.

We used three types of intracellular solution, low chloride (10 mM), medium chloride (30 mM) and high chloride (131.1 mM chloride, the same as in extracellular medium). The 10 mM chloride intracellular solution contained (in mM): 121.1 potassium gluconate, 11 EGTA, 1 $CaCl_2$, 1 $MgCl_2$, 6 TEACl, 2 MgATP, 10 HEPES and 40 sucrose; the pH was adjusted to 7.2 with KOH. The 30 mM chloride intracellular solution contained (in mM): 101.1 potassium gluconate, 11 EGTA, 6 KCl, 1 $CaCl_2$, 1 $MgCl_2$, 20 TEACl, 2 MgATP, 1 HEPES and 40 sucrose; the pH was adjusted to 7.2 with KOH. High chloride intracellular solution contained (in mM): 107.1 KCl, EGTA 11, 1 $CaCl_2$, 1 $MgCl_2$, 10 HEPES, 20 TEACl, 2 MgATP, and 14 sucrose, adjusted to pH 7.2 with KOH. Solutions were prepared with laboratory grade deionised water with a resistivity of 15 MΩ cm. For the pipette solution used in cell-attached single-channel recordings we used high performance liquid chromatography grade (HPLC) water with a resistivity of 18.2 MΩ cm (VWR Chemicals, UK).

### Whole-cell recordings

Whole-cell currents were recorded with an Axopatch 200B amplifier (Molecular Devices, USA). Recordings were pre-

filtered at 5 kHz with the amplifier's built-in 4-pole-low pass Bessel filter, sampled at 20 kHz with a Digidata 1440 A (Molecular Devices) and stored on a computer hard drive. Analysis of traces was performed with Clampfit 10.7 software (Molecular Devices) after additional 1 kHz Gaussian low pass filtering.

Currents were recorded at a nominal holding potential of −40 mV (values corrected for the liquid junction potential would be more negative by 10 mV), except for taurine responses at pH 6.4, where the holding potential was −60 mV to increase the amplitude of the response. Access resistance was never larger than 8 MΩ and was compensated by at least 60% and up to 80%.

A custom-made U-tube tool was used for the application of agonist to the cells (Krishtal & Pidoplichko, 1980). The duration of the agonist applications was controlled manually to last until the agonist current reached a peak (usually ∼1 s). The position of the U-tube was optimised by applying extracellular solution diluted 1:1 with bi-distilled water to the open tip of a recording electrode placed just above the cell and measuring the 20–80% rise time of the current generated by the diluted solution (acceptable values were below 20 ms, typically 2–20 ms; note that kinetic measurements were done in different experiments with piezo-stepper applications to outside-out patches; see below).

Changes in extracellular pH were allowed to equilibrate for 2 min before recording agonist responses. In every cell, a saturating concentration of agonist was applied every 3rd or 4th agonist application, in order to monitor for rundown. If rundown was greater than 30%, the experiment was discarded. In each cell, we obtained a full dose-response curve and a response to a saturating concentration of glycine (10 or 30 mM). Data were fitted with the Hill equation with custom software (CVFIT; https://github.com/DCPROGS), to estimate the maximum peak current ($I_{max}$), the concentration required to elicit 50% of the maximum response ($EC_{50}$) and the Hill coefficient ($n_H$) for every cell. For the concentration-response curves in the figures, responses were normalised to the maximum current in each cell, pooled together and fitted with the Hill equation for display purposes only (parameter estimates from this pooled fit were not used).

## Cell-attached single-channel recordings

Cells were kept in extracellular medium at pH 7.4. Recording pipettes were filled with extracellular solution at the tested pH, acidic (pH 6.4) or alkaline (pH 8.4) with up to 300 mM of agonist. In order to reduce noise level, pipette tips were coated with Sylgard 184 (Dow Corning, Dow Silicones, UK). Currents were recorded with an Axopatch 200B, at a holding potential of +100 mV. Currents were pre-filtered at 10 kHz with the amplifier

built-in 4-pole low-pass Bessel filter and sampled at 100 kHz with a Digidata 1440A. Current traces were filtered with an additional 3 kHz low–pass Gaussian filter and resampled to 33.3 kHz in Clampfit 10.7 software for analysis. Clusters of agonist-elicited openings were accepted for analysis only if they were longer than 100 ms and separated by at least 100 ms shut time.

Open probability ($P_{open}$) for each cluster was determined with the single-channel search protocol in Clampfit 10.7 software, where channel activity is idealised by a half amplitude threshold-crossing method. The $P_{open}$ of a cluster was calculated as the ratio between the total open time and the cluster duration.

## Fast agonist applications

Concentration-jump experiments were performed with a double-barrel application tool made from theta-tube glass (Hilgenberg GmbH, Germany), pulled and cut to a tip with a diameter of ∼150 $\mu$M. The tool was driven by a piezo manipulator (PZ-150M, Burleigh Instruments, UK) and was filled with solutions driven by gravity, with a dead time for solution exchange in the barrels of approximately 2 min.

The exchange time for the agonist pulses at the patch was estimated by applying a pulse of bath solution diluted 1:1 with water to an open electrode tip before the formation of the patch and after its rupture. The 20–80% rise and 80–20% decay times from these currents were measured with Clampfit 10.7 and were on average 150 $\mu$s. Experiments where the rise or decay times were longer than 200 $\mu$s were discarded.

Pulses of glycine were applied every 20 s and pulses of partial agonists ($\beta$-alanine, taurine and GABA) every 10 s.

Macroscopic currents were recorded from outside-out patches with an Axopatch 200B at a nominal pipette holding potential of −100 mV (the actual values, when corrected for junction potential were −102, −110 and −113 mV in experiments with 131.1, 30 and 10 mM intra-cellular chloride, respectively. Analysis was carried out on traces averaged from 5–15 responses. The peak or the responses was measured in Clampfit 10.7 software. Rise time constants were obtained by fitting the response from 20% to 100% of the amplitude with a single exponential. Decay time constants were obtained by fitting decay time from 90% to10% of the peak with one exponential (partial agonists and 2/11 of the glycine experiments at pH 6.4) or two exponentials (most of glycine responses; where fits are bi-exponential we report the weighted time constant, calculated as the average of the two time constants weighted by the fractional area of each component).

## Agonist purification

$\beta$-Alanine, taurine and GABA (Sigma-Fluka, UK; all 300 mM) solutions were tested for glycine contamination

by HPLC assay. Glycine was present in the taurine and GABA samples at 0.5 and 0.4 $\mu$M, respectively. These concentrations are well below to those required to evoke a measurable current response in our recordings (50 $\mu$M at pH 7.4), hence for whole-cell experiments we used the agonists as purchased. However, for single-channel and fast agonist application experiments, we used purified taurine and GABA, obtained by repeated crystallisation from aqueous ethanol. Glycine removal was confirmed by re-testing the purified taurine and GABA samples.

## Statistics

Results are reported as the mean $\pm$ SD and *n* is given for the number of cells, clusters or patches as indicated. When two groups were compared, data were tested for normality and significance was tested as appropriate with unpaired or paired Student's *t* test (whole-cell recordings and fast agonist applications, respectively) to obtain the *P* values reported.

For non-parametric data (such as open probability measurements) we used a randomisation test (two-tail, non-paired; 10,000 iterations) to determine *P* values for the difference being greater than or equal to the observed difference (DC-Stats software: https://github.com/DCPROGS/DCSTATS/releases/tag/v.0.3.1-alpha). Given multiple comparisons were sometimes necessary, differences are to be considered statistically significant for $P < (0.05/3)$, where 3 is the Bonferroni correction for three comparisons.

## Results

### Acidification of extracellular pH reduces GlyR agonist potency and the maximum response to partial agonists relative to glycine

Figure 1*A* and 1*C* show how GlyR agonist responses are affected by changing the extracellular pH by one unit from its physiological value of 7.4 (Fig. 1*B*). Traces in the top panels are whole-cell responses elicited by the U-tube application of agonists to zebrafish WT $\alpha$1 GlyR expressed in HEK 293 cells. The concentration-response curves thus

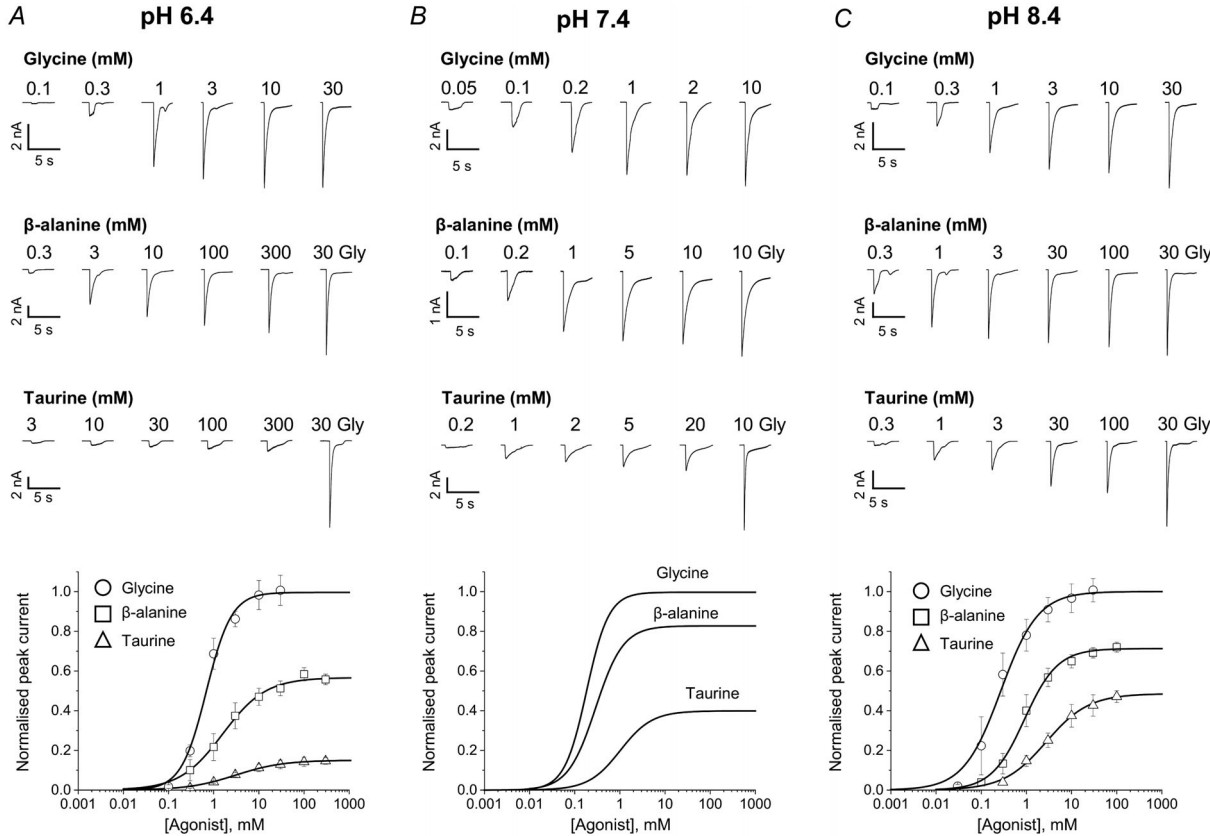

**Figure 1. Acidic pH reduces agonist potency and efficacy measured from whole-cell GlyR responses**
Upper panels, whole-cell current responses to U-tube agonist applications to HEK 293 cells expressing zebrafish $\alpha$1 GlyR in acidic pH 6.4 (*A*), physiological pH 7.4 (*B*) and alkaline pH 8.4 (*C*). Lower panels, averaged concentration-response curves to glycine (open circles), $\beta$-alanine (open squares) and taurine (open triangles) (*n* = 5–10; see Table 1; responses are normalised to those to saturating glycine concentrations, 10 or 30 mM, in each cell; error bars show SD values). Data at pH 7.4 are from Ivica *et al*. (2021).

**Table 1. Effect of pH on the dose-response curves to GlyR agonists obtained in whole-cell recordings of zebrafish α1 GlyR**

| | | Glycine | β-Alanine | Taurine |
|---|---|---|---|---|
| pH 6.4 | $EC_{50}$ ($\mu$M) | 680 ± 110 | 1890 ± 750 | 3200 ± 700 |
| | $nH$ | 1.61 ± 0.30 | 0.93 ± 0.15 | 0.87 ± 0.13 |
| | $I_{max}$ (nA) | 7.5 ± 2.3 | 10.5 ± 4.2 | 12.9 ± 2.6 |
| | $I_{rel}$ | | 0.57 ± 0.17 | 0.15 ± 0.12 |
| | $n$ (cells) | 7 | 9 | 5 |
| pH 7.4 | $EC_{50}$ ($\mu$M) | 190 ± 60 | 340 ± 190 | 1050 ± 220 |
| | $nH$ | 1.9 ± 0.3 | 1.34 ± 0.19 | 1.26 ± 0.12 |
| | $I_{max}$ (nA) | 8.6 ± 1.2 | 7.0 ± 3.0 | 9.8 ± 3.1 |
| | $I_{rel}$ | | 0.84 ± 0.09 | 0.40 ± 0.20 |
| | $n$ (cells) | 8 | 10 | 6 |
| pH 8.4 | $EC_{50}$ ($\mu$M) | 280 ± 110 | 990 ± 260 | 3400 ± 1400 |
| | $nH$ | 1.22 ± 0.3 | 1.14 ± 0.15 | 0.90 ± 0.16 |
| | $I_{max}$ (nA) | 7.0 ± 1.7 | 8.7 ± 2.7 | 8.8 ± 2.5 |
| | $I_{rel}$ | | 0.74 ± 0.09 | 0.50 ± 0.11 |
| | $n$ (cells) | 7 | 6 | 5 |

Agonist maximum currents and $EC_{50}$ values are from fits of the Hill equation and are shown as mean ± SD. Data at pH 7.4 are from Ivica *et al.* (2021). $I_{rel}$ is the maximum response normalised to that of glycine in the same cell.

obtained (normalised to the glycine maximum in each cell) are shown in the bottom panels.

Reducing extracellular pH from 7.4 to 6.4 had two clear effects (Fig. 1*A* and Table 1). First, it decreased receptor sensitivity for all agonists. $EC_{50}$ values increased by 3.6-fold for the full agonist glycine (from 190 $\mu$M to 680 $\mu$M; $n = 7$ cells; $P << 10^{-6}$) and by 5.6-and 3-fold for the partial agonists β-alanine and taurine (to 1890 and 3200 $\mu$M from 340 and 1050 $\mu$M, $n = 9$ and 5 cells, $P = 0.00002$ and 0.0003, respectively). Secondly, acidification reduced the maximum responses to partial agonists, relative to that of the full agonist glycine. This decrease was marked for β-alanine, from 84% at pH 7.4 to 57% at pH 6.4 ($n = 9$ cells, $P = 0.00004$) and even greater for taurine, with a 2.7-fold decrease from 40% to 15% ($n = 5$ cells, $P = 0.006$).

The effects of increasing pH by one unit were similar to those of acidification, but much smaller (see Fig. 1*C* and Table 1). The only change that was large and consistent enough to reach statistical significance was the decrease in the potency of the partial agonists β-alanine and taurine (about 3-fold, from 340 and 1050 $\mu$M at pH 7.4 to 990 and 3400 $\mu$M at pH 8.4, respectively; $n = 6$ cells, $P = 0.0001$; $n = 5$ cells, $P = 0.0027$, unpaired *t* test; cf. from 190 to 280 $\mu$M for glycine, $n = 8$ cells, $P = 0.09$).

### Single-channel recordings at high agonist concentrations confirm that agonist efficacy is lower at pH 6.4

The whole-cell data, particularly the reductions in the relative maximum of partial agonist responses, suggest that acidification reduces agonist efficacy. However, macroscopic responses may be distorted by changes in desensitisation and do not allow the measurement of agonist efficacy as an absolute value. In addition to that, in whole-cell experiments, all the cell is exposed to the pH changes, which could elicit membrane currents *per se*. A much more robust alternative way of measuring agonist efficacy in ligand-gated channels is by single-channel recording at high agonist concentration.

Continuous cell-attached traces from this type of experiment are shown at the top of Fig. 2, where the upward deflections in the trace are GlyR openings in response to 30 mM glycine in the pipette.

In the presence of high agonist concentrations, GlyRs in the patch are desensitised most of the time (dashed lines in Fig. 2*A*). From time to time a channel emerges from desensitisation and then it produces a cluster of openings and closings, before entering a long-lived desensitised state again. During cluster activity, we can be sure that only one GlyR is active (because there are no double amplitude openings) and thus we can measure the maximum probability of a single GlyR opening in response to an agonist, as $P_{open}$, the open time in a cluster expressed as a fraction of the total cluster duration. This analysis excises periods of desensitisation (between clusters).

The single-channel traces from Fig. 2 show zebrafish α1 GlyR responses to agonists at three different pH values, 6.4, 7.4 and 8.4 (panels *A*, *B* and *C*, respectively). The single-channel data strongly support the conclusion from the whole-cell experiments that agonist efficacy is lower at pH 6.4 and that it is not substantially affected by increasing the pH to 8.4. The top trace in Fig. 2 shows three clusters of openings evoked by glycine in a continuous recording

**Table 2. Single-channel parameters measured for zebrafish α1 GlyR in three extracellular pH conditions**

| pH | | Glycine | β-Alanine | Taurine |
|---|---|---|---|---|
| 6.4 | Max $P_{open}$ | 0.93 ± 0.11 | 0.57 ± 0.24 | 0.34 ± 0.20 |
| | $n_{patches}$ ($n_{clusters}$) | 6 (46) | 11 (55) | 7 (49) |
| | Agonist concentration (mM) | 30 | 300 | 300 |
| 7.4 | Max $P_{open}$ | 0.97 ± 0.05 | 0.91 ± 0.21 | 0.66 ± 0.24 |
| | $n_{patches}$ ($n_{clusters}$) | 10 (48) | 7 (30) | 7 (71) |
| | Agonist concentration (mM) | 10 | 30 | 100 |
| 8.4 | Max $P_{open}$ | 0.96 ± 0.05 | 0.81 ± 0.18 | 0.62 ± 0.23 |
| | $n_{patches}$ ($n_{clusters}$) | 9 (35) | 11 (43) | 5 (47) |
| | Agonist concentration (mM) | 30 | 100 | 300 |

Values are shown as mean ± SD. Data at pH 7.4 are from Ivica *et al.* 2021.

at pH 6.4. Comparison of these clusters with those at 7.4 (panel *B*) clearly shows that even for the highly efficacious agonist glycine, open probability is higher at physiological pH, where the clusters appear 'whiter', as they have fewer short shuttings.

This is confirmed by the boxplots of the $P_{open}$ values from each cluster (Fig. 3*A*; Table 2). Glycine is very efficacious at both pH values, but slightly less so at acidic pH, than at physiological pH (0.93 cf. 0.97; *n* = 30 clusters, *P* = 0.0012, two-tailed randomisation test). The

greater single-channel open probability at pH 7.4 is more obvious for the partial agonists taurine and β-alanine. The maximum $P_{open}$ elicited by taurine decreased about 2-fold, from 0.66 at pH 7.4 to 0.34 at pH 6.4 (*n* = 49 clusters, $P << 10^{-6}$; Fig. 3*A*, Table 2). β-Alanine is close to being a full agonist at pH 7.4, with a maximum $P_{open}$ of 0.91, but is clearly partial at pH 6.4, where the two clusters in Fig. 2*A* had a $P_{open}$ of 0.59 and 0.47 (maximum $P_{open}$ 0.57, *n* = 57 clusters, $P < 10^{-6}$). These experiments also confirm that increasing pH to 8.4 had negligible effects

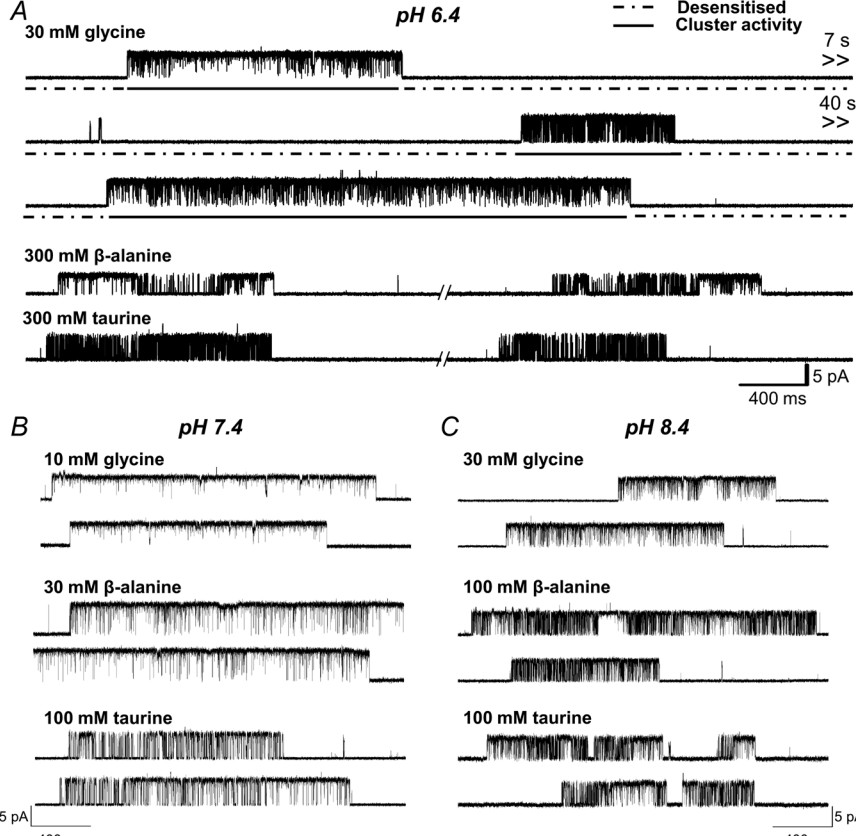

**Figure 2. Acidic pH decreases agonist efficacy in zebrafish α1 GlyR**
Representative single-channel traces were recorded in the cell-attached configuration at saturating agonist concentrations at pH 6.4, 7.4 and 8.4 (panels *A*, *B* and *C*, respectively; upward deflections show channel openings, currents are inward and are caused by chloride exit).

on agonist efficacy. At 8.4, maximum $P_{open}$ values were 0.96, 0.81 and 0.62 for glycine, $\beta$-alanine and taurine (*vs.* 0.97, 0.91 and 0.66 at pH 7.4; $P = 0.1836$, 0.047 and 0.391, respectively).

The decrease in maximum $P_{open}$ at acidic pH was associated with an increase in the range of the $P_{open}$ values for the more efficacious agonists, as their average maximum $P_{open}$ moved away from 1 (see the interquartile boxes for glycine and $\beta$-alanine, Fig. 3*A*). The traces in Fig. 2*A* show an example of clusters with different $P_{open}$ values for glycine at pH 6.4, 0.99 and 0.95 in the first and third cluster and 0.76 in the second cluster. This effect was not obvious for taurine, whose control value of $P_{open}$ (0.66) is close to 0.5, the value where $P_{open}$ variability would be expected to be greatest.

The amplitude of single-channel currents varied slightly from patch to patch. As these recordings were obtained in the cell-attached configuration in normal extracellular solution, their true holding voltage is unknown, as it will be the imposed voltage plus the (unknown) cell resting potential. However, acidic pH increases the conductance of some GABA receptors (Mortensen *et al.* 2010; but see Kisiel *et al.* 2019). We found that GlyR single-channel conductance was not affected by pH, as the slope of current amplitude-voltage plots was similar at pH 6.4 and

7.4, (e.g. respectively $71.9 \pm 2.9$ and $73.8 \pm 6.4$ pS, $n = 4$ patches, $P = 0.6114$, unpaired $t$ test; Fig. 3*B* and *C*).

## Effect of pH acidification on responses to fast agonist applications: symmetrical chloride

Given the marked effect of extracellular acidification on agonist efficacy and potency, we next examined how it affected the kinetics of receptor activation and deactivation, by recording macroscopic responses to fast agonist applications to outside-out patches. These experiments should help us identify which step in the receptor activation mechanism is involved and establish whether pH change can affect glycinergic synaptic currents. The agonist pulses (with a piezo-driven theta tube, 3 mM; 2 ms; exchange time constant of 150 $\mu$s) mimic the time course of glycine at the synaptic cleft, where glycine reaches 2.2–3.5 mM, but is cleared within 1 ms (Beato, 2008). In the first instance we carried out our experiments in high symmetrical chloride (e.g. filling recording pipettes with a high intracellular chloride solution, 131.1 mM), in order to optimise the sensitivity of our experiments, by maximising the size of the responses and minimising possible chloride gradient shifts artefacts (Karlsson *et al.* 2011).

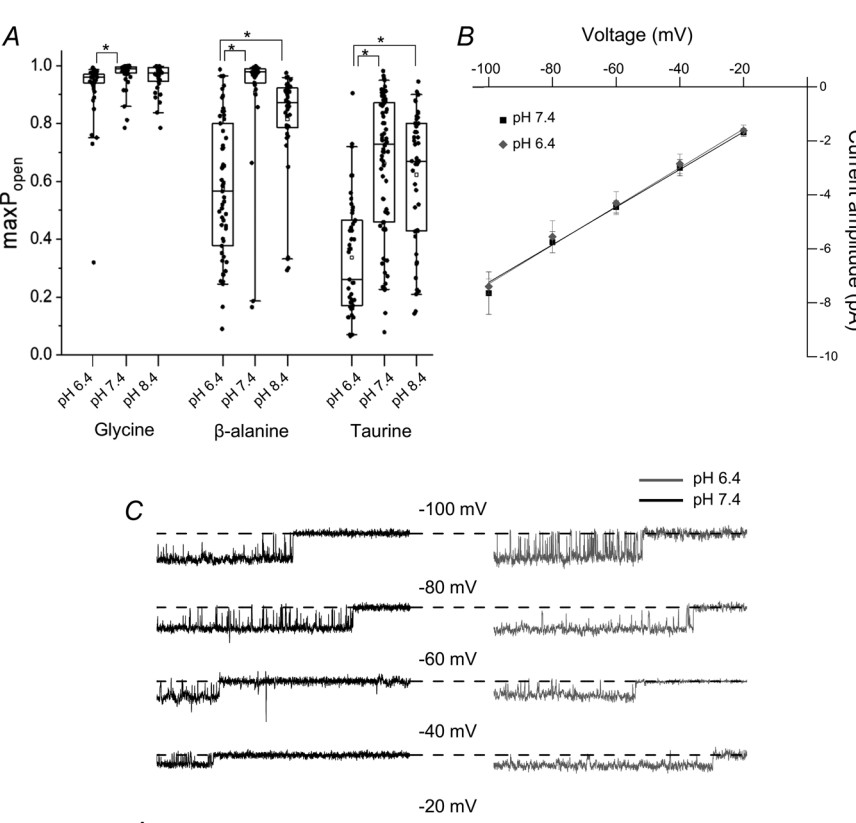

**Figure 3. Acidic pH decreases agonist efficacy but does not affect the conductance of zebrafish $\alpha$1 GlyR**
*A*, cluster $P_{open}$ values produced by saturating agonist concentrations at pH 6.4, 7.4 and 8.4; data at pH 7.4 are from Ivica *et al.* (2021). Boxes and whiskers show the 25th and 75th and the 5th and 95th percentiles, respectively. The horizontal line in the box is the median. Asterisked brackets denote significance; see text for P values. *B*, current–voltage relationship for $\alpha$1 GlyR single-channel openings at pH 6.4 (grey diamonds) and at pH 7.4 (black squares) recorded from outside-out patches ($n = 4$ patches, error bars represent SD; the black and the grey lines are fits to the data pooled from four patches at pH 7.4 and 6.4, respectively). *C*, traces are outside-out recordings from $\alpha$1 GlyR in the presence of 10 mM glycine (symmetrical 131.1 mM chloride). Openings are downward and are inward currents, dashed lines indicate closed level

**Table 3. Effect of extracellular acidification on the amplitude and time course of currents of zebrafish $\alpha 1$ GlyR elicited by concentration jumps with glycine, $\beta$-alanine, taurine and GABA. The measurements were done with 131.1 mM intracellular chloride at $-100$ mV. Agonist concentrations were 3 mM (glycine), 50 mM ($\beta$-alanine at pH 7.4) and 100 mM for taurine and GABA and for $\beta$-alanine at pH 6.4**

| Agonist ($n$ patches) | Glycine (6) | | $\beta$-Alanine (6) | | Taurine (6) | | GABA (5) | |
|---|---|---|---|---|---|---|---|---|
| | pH 7.4 | pH 6.4 | pH 7.4 | pH 6.4 | pH 7.4 | pH 6.4 | pH 7.4 | pH 6.4 |
| Peak current amplitude (pA) | $1160 \pm 630$ | $970 \pm 520$ $P = 0.016$ | $910 \pm 280$ | $630 \pm 210$ $P = 0.019$ | $970 \pm 225$ | $570 \pm 180$ $P = 0.010$ | $1170 \pm 430$ | $480 \pm 97$ $P = 0.014$ |
| Peak current amplitude (% of pH 7.4 value) | | $83 \pm 8\%$ | | $72 \pm 16\%$ | | $61 \pm 20\%$ | | $44 \pm 15\%$ |
| Decay $\tau$ (ms) | $25.5 \pm 8.1$ | $9.7 \pm 4.3$ $P = 0.012$ | $5.7 \pm 2.0$ | $2.6 \pm 0.8$ $P = 0.002$ | $4.3 \pm 1.1$ | $2.3 \pm 0.7$ $P = 0.006$ | $2.5 \pm 0.5$ | $1.3 \pm 0.3$ $P = 0.002$ |
| Rise time $\tau$ (ms) | $0.17 \pm 0.07$ | $0.29 \pm 0.12$ $P = 0.037$ | $0.19 \pm 0.04$ | $0.35 \pm 0.10$ $P = 0.003$ | $0.23 \pm 0.05$ | $0.40 \pm 0.10$ $P = 0.003$ | $0.35 \pm 0.06$ | $0.57 \pm 0.18$ $P = 0.02$ |

$P$ values for comparison with pH 7.4; paired sample $t$ test.

pH 6.4 (grey traces in Fig. 4$A$) had two effects (cf. black traces, control responses at pH 7.4 in the same patch). The first was a reduction in the amplitude of all agonist evoked currents (Table 3). This decrease was modest for glycine (to $83 \pm 8\%$ of the current at pH 7.4; $n = 6$ patches, $P = 0.016$, paired $t$ test) and much more marked for partial agonists such as $\beta$-alanine, taurine and GABA, where currents decreased to $72 \pm 16\%$, $61 \pm 20\%$ and $44 \pm 15\%$ of their pH 7.4 controls ($n = 6$, 6 and 5 patches, $P = 0.019$, 0.010 and 0.014, respectively; paired $t$ test; Fig. 4$B$, Table 3).

Acidification also changed the time course of the responses. This is shown by the second pair of traces in each panel of Fig. 4$A$, where currents at pH 6.4 are scaled to their peak in control pH. The speed of the current decay after the end of the agonist application was found to be inversely correlated with efficacy and was slowest for glycine (note the different time scale of the glycine traces). At low pH, current decay became approximately 2-fold faster for all agonists and this effect was clear, despite interpatch variability (Fig. 4$C$ and Table 3; see Papke & Grosman, 2014). For glycine, the weighted decay time

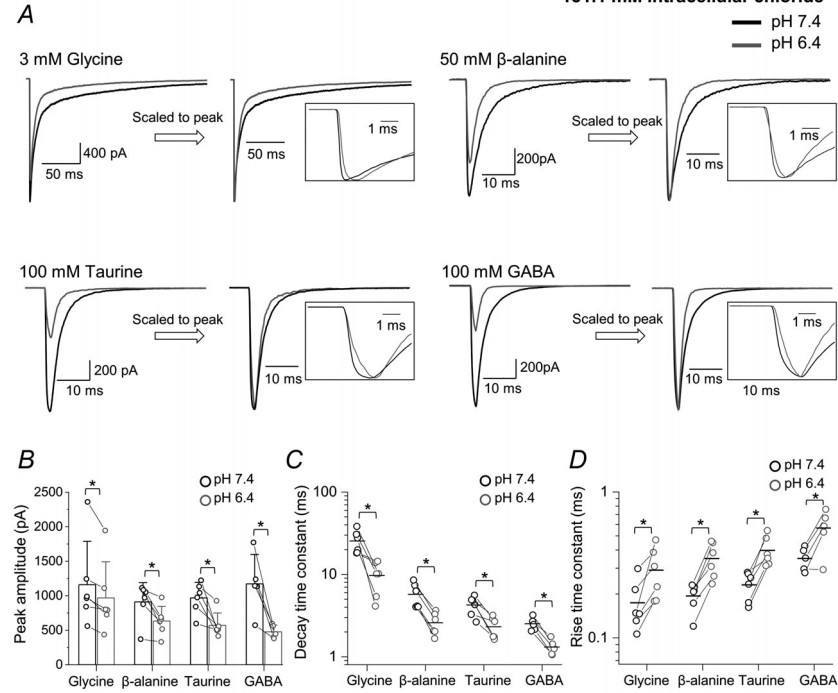

**Figure 4. Acidification to pH 6.4 reduces the amplitude and changes the time course of outside-out responses of zebrafish $\alpha 1$ GlyR to fast agonist applications**
$A$, representative inward currents elicited by 2 ms pulses of glycine, $\beta$-alanine, taurine and GABA, at pH 7.4 (black) or 6.4 (grey). Traces are averages of at least five current responses and patches were held at $-100$ mV. In the right panels the responses are scaled to their peak to allow comparison of current kinetics (see also expanded insets). $B$, bar chart presenting the peak amplitudes of concentration pulses measured for glycine, alanine, taurine and GABA; error bars show SD, lines connect values from the same patch. $C$ and $D$, decay and rise time constants, respectively, for currents elicited by the four agonists at pH 7.4 (black) and 6.4 (grey). Horizontal lines are means and lines connect values from the same patch. Asterisked brackets in $B$, $C$ and $D$ show comparisons that were statistically significant; see Table 3 for $P$ values.

constant (see Methods) decreased from $25.5 \pm 8.1$ ms at pH 7.4 to $9.7 \pm 4.3$ ms at pH 6.4 ($n = 6$ patches, $P = 0.012$, paired $t$ test). The corresponding values for $\beta$-alanine, taurine and GABA were $5.7 \pm 2.0$, $4.3 \pm 1.1$ and $2.5 \pm 0.5$ ms at pH 7.4 and $2.6 \pm 0.8$, $2.3 \pm 0.05$ and $1.3 \pm 0.3$ ms at pH 6.4 ($n = 6$, 6 and 5 patches; $P = 0.002$, 0.006 and 0.002, respectively; paired $t$ test).

The current rise time became detectably slower in acidic pH for all agonists (Table 3; Fig. 4*A*, inset traces). For glycine, this time constant increased from $0.17 \pm 0.07$ ms (pH 7.4) to $0.29 \pm 0.12$ ms at acidic pH 6.4 ($n = 6$ patches, $P = 0.037$, paired $t$ test). Note that with an exchange time of 0.15 ms, this is not a meaningful measurement for glycine at physiological pH, as the rate of onset of the response is limited by the rate of agonist exchange, rather than the receptor mechanism. Thus, while we are sure that receptor activation by glycine slows down in low pH, we do not know by how much.

This was also true for $\beta$-alanine, where the rise time $\tau$ increased from $0.19 \pm 0.04$ ms at pH 7.4 to $0.35 \pm 0.10$ ms at pH 6.4 ($n = 6$ patches, $P = 0.003$, paired $t$ test). Rise times for the partial agonists taurine and GABA were slower at pH 7.4 and became even slower at pH 6.4, with $\tau$ values increasing from $0.23 \pm 0.05$ to $0.40 \pm 0.10$ ms for taurine ($n = 6$ patches, $P = 0.003$, paired $t$ test) and from $0.35 \pm 0.10$ ms to $0.57 \pm 0.18$ ms for GABA ($n = 5$ patches, $P = 0.02$, paired $t$ test; Fig. 4*D*, Table 3).

### Effect of pH acidification on responses to fast agonist applications: lower intracellular chloride

The symmetrical chloride conditions used above are not physiological, as the intracellular chloride concentration in adult neurones is likely to be in the low millimolar

range (Doyon *et al.* 2016). Given that high intracellular chloride slows deactivation of GlyR (Pitt *et al.* 2008), it is important to establish whether the effects of pH acidification reported above are also present in lower internal chloride, a condition more relevant to physiological conditions and to the cell-attached single-channel recordings. We started with repeating our experiments at 30 mM intracellular chloride (a compromise condition where responses are still large) and tested glycine and taurine.

Figure 5 shows the time course of the effect of pH 6.4 on consecutive current responses to glycine pulses (2 ms, 30 mM, 1 every 20 s). Both control and agonist solutions were changed to the new pH at the time marked by the arrows and it took approximately 100 s for the solution exchange in the perfusion system to be complete. In this patch, pH 6.4 roughly halved the response peak amplitude from 827 pA to 447 pA (average of five responses, responses 1–5 at 7.4 and 11–15 at pH 6.4). Figure 5 also shows that effect was reversible and returning the patch to pH 7.4 restored the peak amplitude to 770 pA (e.g. 93% of the control value). In the 11 patches in which glycine was tested, its peak response was decreased on average to $64 \pm 22\%$ of its control value ($n = 11$ patches, $P = 0.002$; Table 4). In 7/11 patches, the recording was long enough for us to measure the current recovery, to $95 \pm 10\%$ (Fig. 5*C*). A greater effect of acidic pH was observed for taurine responses (2 ms, 100 mM), where the peak amplitude decreased to $32 \pm 5\%$ of control ($n = 7$ patches; from $354 \pm 183$ pA to $112 \pm 55$ pA, $P = 0.003$).

The effects of acidic pH on the time course of the responses were also maintained with 30 mM intracellular chloride. The decay of agonist currents became faster at low pH for both glycine (from $17.8 \pm 10.6$ ms to $8.1 \pm 3.9$ ms, weighted time constant values, see Methods;

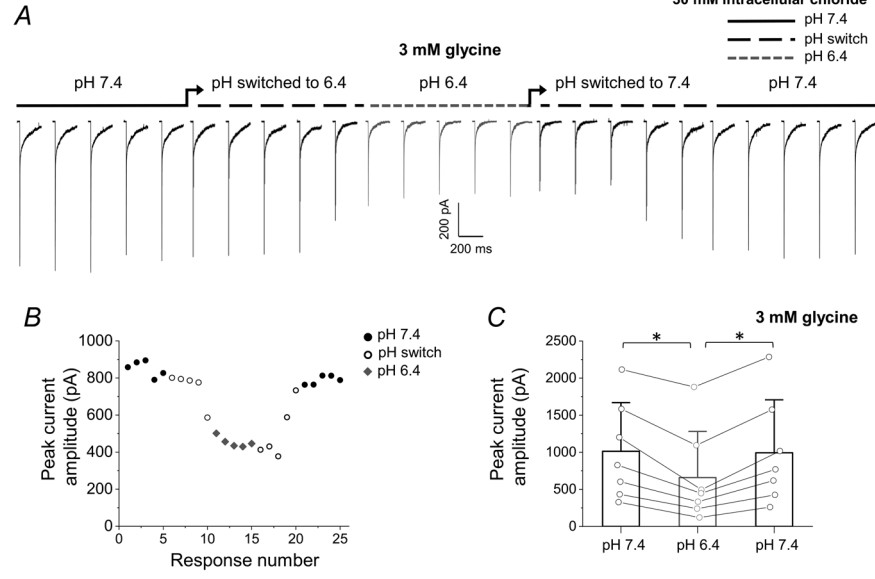

**Figure 5. Acidic pH reversibly reduces the amplitude of glycine responses of zebrafish α1 GlyR in 30 mM intracellular chloride**

*A*, consecutive current responses recorded from the same outside-out patch, held at −100 mV, in response to 3 mM 2 ms glycine pulses at pH 7.4 and after pH change to 6.4 (of both the agonist and the control solution; grey traces) and back to 7.4. Responses obtained during the solution exchange in the application tool are marked by a black dashed line above the traces and were not analysed. *B*, plot of the peak amplitude of the responses shown in the same patch. *C*, peak amplitudes of glycine responses in patches with recovery. Asterisked brackets denote significance, $P = 0.0025$ and $P = 0.001$ when amplitude at pH 6.4 is compared to the preceding or the following pH 7.4 control.

**Table 4. Effect of acidic pH on the amplitude and time course of currents elicited by 2 ms 3 mM glycine concentration jumps mM on homomeric GlyR (zebrafish α1) and on mammalian heteromeric GlyR (rat α1β)**

| | Zebrafish α1 GlyR | | | | | |
|---|---|---|---|---|---|---|
| Intracellular chloride (mM) (*n* patches) | 131.1 (6) | | 30 (11) | | 10 (7) | |
| | pH 7.4 | pH 6.4 | pH 7.4 | pH 6.4 | pH 7.4 | pH 6.4 |
| Peak current (pA) | 1160 ± 630 | 970 ± 520 | 746 ± 620 | 498 ± 540 | 312 ± 375 | 199 ± 259 |
| | | P = 0.016 | | P = 0.002 | | P = 0.046 |
| Peak current (% of pH 7.4 value) | | 83 ± 8% | | 64 ± 22% | | 60 ± 21% |
| Decay τ (ms) | 25.5 ± 8.1 | 9.7 ± 4.3 | 17.8 ± 10.6 | 8.1 ± 3.9 | 10.6 ± 2.5 | 4.5 ± 2.4 |
| | | P = 0.012 | | P = 0.004 | | P = 0.0001 |
| Rise time τ (ms) | 0.17 ± 0.07 | 0.29 ± 0.12 | 0.24 ± 0.06 | 0.44 ± 0.18 | 0.26 ± 0.09 | 0.38 ± 0.18 |
| | | P = 0.037 | | P = 0.001 | | P = 0.186 |

| | Rat α1β GlyR | | | | | |
|---|---|---|---|---|---|---|
| Intracellular chloride (mM) (*n* patches) | 131.1 (7) | | 30 (8) | | 10 (7) | |
| | pH 7.4 | pH 6.4 | pH 7.4 | pH 6.4 | pH 7.4 | pH 6.4 |
| Peak current (pA) | 348 ± 200 | 318 ± 200 | 365 ± 364 | 290 ± 283 | 106 ± 48 | 79 ± 35 |
| | | P = 0.013 | | P = 0.04 | | P = 0.003 |
| Peak current (% of pH 7.4 value) | | 90 ± 9% | | 78 ± 6% | | 73 ± 8% |
| Decay τ (ms) | 16.5 ± 8.8 | 10.5 ± 8.5 | 10.4 ± 3.7 | 6.4 ± 1.5 | 7.6 ± 2.2 | 4.5 ± 1.4 |
| | | P = 0.001 | | P = 0.01 | | P = 0.013 |
| Rise time τ (ms) | 0.20 ± 0.06 | 0.27 ± 0.08 | 0.27 ± 0.13 | 0.37 ± 0.18 | 0.22 ± 0.06 | 0.32 ± 0.08 |
| | | P = 0.001 | | P = 0.004 | | P = 0.002 |

The measurements were done with 131.1 mM intracellular chloride (e.g. symmetrical), 30 mM and 10 mM intracellular chloride; nominal holding potential −100 mV (effective holding potential −102, −110 and −113 mV when corrected for junction potential with 131.1, 30 and 10 mM intracellular chloride).

*n* = 11 patches *P* = 0.004, paired *t* test; Fig. 6*A* and *C*; Table 4) and taurine (from 2.5 ± 0.8 ms to 1.5 ± 0.6 ms at pH 6.4; *n* = 7 patches, *P* = 0.002; Fig. 6*B* and *C*).

GlyR activation became at least 2-fold slower at acidic pH for both glycine and taurine, increasing from 0.24 ± 0.06 ms to 0.44 ± 0.18 ms for glycine and from 0.24 ± 0.07 to 0.44 ± 0.11 ms for taurine (glycine: *n* = 11 patches, *P* = 0.001, paired *t* test; Fig. 6*D*; Table 4; taurine: *n* = 7 patches, *P* = 0.0003, paired *t* test, Fig. 6*D*).

Given that the effect of acidic pH was maintained with 30 mM intracellular chloride, we extended our experiments, decreasing intracellular chloride to 10 mM and recording currents produced from heteromeric GlyRs, which are likely to be the synaptic channels in the adult mammalian CNS.

The traces in Fig. 7 confirm that the loss of GlyR function with acidification persisted at physiological intracellular chloride and in heteromeric receptors. Predictably (Table 4), the responses recorded with 10 mM intracellular chloride were much smaller and obtaining measurable recordings became more challenging: for both receptor types, the peak amplitude decreased by 2.5- to 3-fold cf. symmetrical 131.1 mM chloride.

Even with these smaller responses, the main features of the acidification effects were consistent: the peak amplitudes decreased (by 40 ± 21% and 27 ± 8% for homomeric and heteromeric receptors, *n* = 7 and 7, Table 4), the decay time constants became faster (from 10.6 ± 2.5 to 4.5 ± 2.4 ms and from 7.6 ± 2.2 to 4.5 ± 1.4 ms) and the rise time constants became slower (from 0.26 ± 0.09 to 0.38 ± 0.18 ms and from 0.22 ± 0.06 to 0.32 ± 0.08 ms).

The consistent message from these experiments was that at pH 6.4 responses to fast synaptic-like glycine applications became smaller in amplitude, slower in onset and faster in decay, irrespective of the GlyR isoform and the intracellular chloride concentration.

## Discussion

Our results show that the maximum open probability produced by full or partial agonists on homomeric α1 GlyR is reduced by extracellular acidification from pH 7.4 to 6.4, because of a decrease in efficacy. In the same experiments, little or no effect was observed when pH was

increased by one unit. Acidic extracellular conditions also affected the amplitude and the time course of macroscopic agonist currents designed to replicate synaptic conditions. These changes were the same at a range of intracellular chloride concentrations (10 to 131.1 mM), suggesting that acidification by one unit is likely to impair signalling at glycinergic synapses and that this effect is not an artefact of chloride gradient rundown.

## Acidic pH reduces gating efficacy to a similar extent for all GlyR agonists

In whole-cell dose-response curve experiments (Fig. 1 and Table 1), decreasing extracellular pH by one unit reduced GlyR sensitivity to the agonists glycine, $\beta$-alanine and taurine, roughly by the same amount, 3- to 5.6-fold. This confirms previous reports for glycine on native and

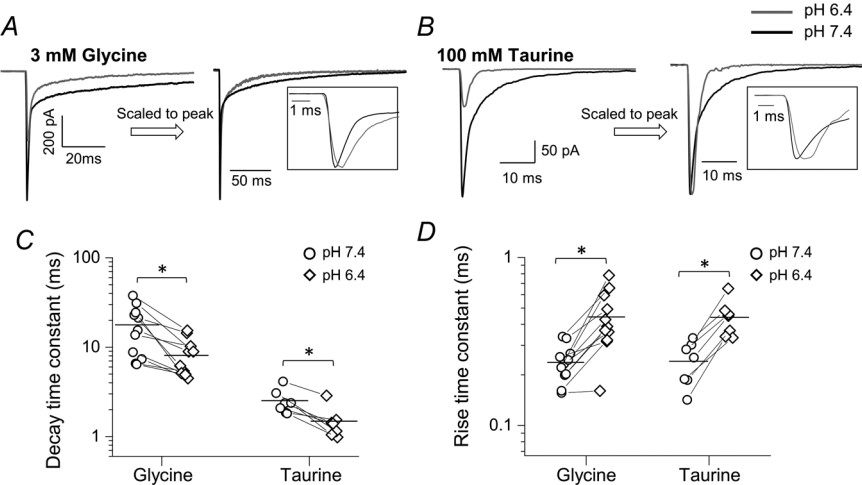

**Figure 6. The effects of acidic pH on the time course of fast agonist responses of zebrafish $\alpha$1 GlyR (30 mM intracellular chloride)**
Representative macroscopic inward currents elicited by 2 ms concentration pulses of glycine (*A*) and taurine (*B*) from outside-out patches held at −100 mV (−110 mV when corrected for junction potential). Traces are averages of at least five current responses. In the right panels the responses are scaled to their own peak to allow comparison of current kinetics; see also insets. *C* and *D*, decay and rise time constants, respectively, for glycine and taurine responses at pH 6.4 (diamonds) and 7.4 (circles). Horizontal line are means and lines connect the measurements from the same patch. Asterisked brackets in *C* and *D* denote significance, in paired *t* tests; for *P* values see text and Table 4.

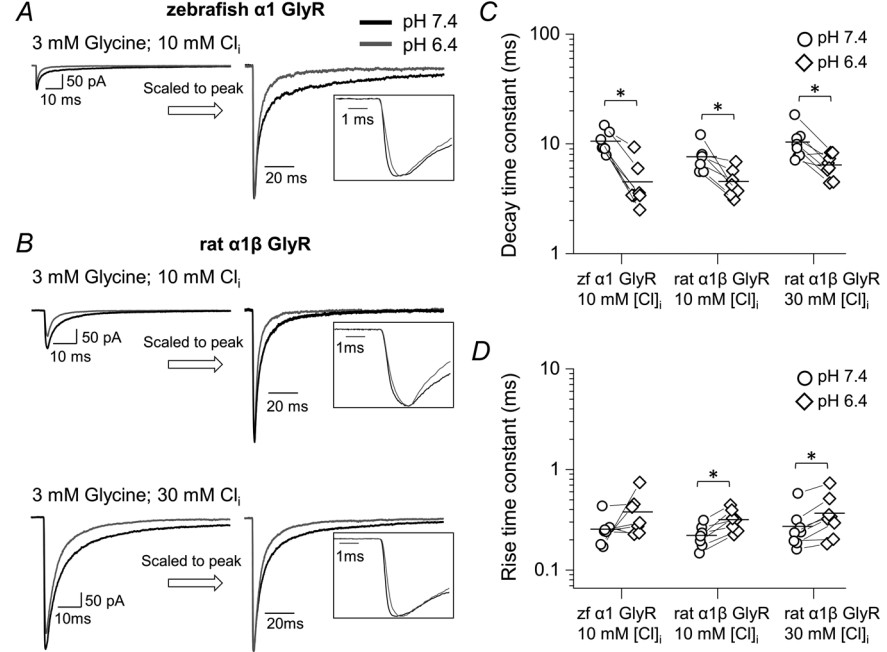

**Figure 7. The effects of acidic pH on responses relevant to native synaptic currents on zebrafish $\alpha$1 GlyR and rat $\alpha$1$\beta$ GlyR 10 mM intracellular chloride**
The panels on the left show macroscopic inward currents elicited by 3 mM, 2 ms concentration pulses of glycine on homomeric zebrafish $\alpha$1 GlyR (*A*) and heteromeric rat $\alpha$1$\beta$ GlyR (*B*) recorded in outside-out patches held at a nominal potential of −100 mV (actual value, −110 and −113 mV for experiments with 30 and 10 mM intracellular chloride, respectively). Traces are averages of at least five current responses. *C* and *D*, decay and rise time constants, respectively, for glycine and taurine responses at pH 6.4 (diamonds) and 7.4 (circles). Horizontal lines are means and lines connect the measurements from the same patch. Asterisked brackets in *C* and *D* denote significance, in paired *t* tests, for *P* values see text and Table 4.

recombinant receptors (Li *et al.* 2003; Chen *et al.* 2004). In native neurones, a parallel shift of the dose-response curve to higher concentrations was attributed to an effect of protons on the receptor affinity for glycine (Li *et al.* 2003), but no effect of pH on maximum responses has been described. Our results show for the first time that acidic pH also causes a decrease in the maximum whole-cell response to the partial agonists $\beta$-alanine and taurine relative to glycine.

This effect on partial agonist suggests that acidic pH impairs channel gating (Colquhoun, 1998), a conclusion robustly confirmed by single-channel recordings, where we observed a marked fall in the maximum $P_{open}$ for $\beta$-alanine and taurine (from 0.91 and 0.66 to 0.57 and 0.34 at pH 7.4 and 6.4, respectively). A consistent, but very much smaller effect was also observed for glycine, whose maximum $P_{open}$ decreased from 0.97 to 0.93. Despite the big difference between agonists in the magnitude of this effect, these numbers are likely to reflect a similar decrease of about 3-fold in the equilibrium constant that describes efficacy for the three agonists.

Maximum open probability in the absence of desensitisation, as measured here, is related to the overall gating equilibrium constant $E_{eff}$ by the equation:

$$\max P_{open} = \frac{E_{eff}}{E_{eff} + 1}, \qquad (1)$$

where $E_{eff}$ is a function of both the transition to a pre-opening intermediate ('flipping'; Burzomato *et al.* 2004) and the channel pore opening (and their equilibrium constants $F$ and $E$, respectively)

$$E_{eff} = \frac{EF}{F + 1} . \qquad (2)$$

Glycine is a full agonist of the GlyR and its 0.97 maximum $P_{open}$ reflects an $E_{eff}$ value between 28 and 40 (no data scatter taken into account for simplicity in this rough calculation). This range of values agrees with our estimate of an $E_{eff}$ of 34 from global kinetic fitting of mammalian homomeric GlyR data (where fully liganded $E = 38$ and $F = 8$; Burzomato *et al.* 2004). The homomeric zebrafish receptor is likely to be somewhat easier to open than the mammalian receptor (see Ivica *et al.* 2021) and thus to have an $E_{eff}$ for glycine at the top of the range. If we start with an $E_{eff}$ of 40, we need a 3-fold decrease in $E_{eff}$ to reduce maximum $P_{open}$ to glycine from 0.97 to 0.93. At pH 7.4, $\beta$-alanine and taurine elicit a maximum $P_{open}$ of 0.91 and 0.67, reflecting $E_{eff}$ values of approximately 10 and 2, respectively. If pH 6.4 decreases $E_{eff}$ for all agonists by 3-fold, we predict maximum $P_{open}$ values of 0.77 and 0.40 for $\beta$-alanine and taurine, respectively, values that are reasonably close to the observed ones ($0.57 \pm 0.24$ and $0.34 \pm 0.20$; Table 2). As the overall gating constant $E_{eff}$ is linearly related to the opening equilibrium constant $E$ (eqn (2)) and as $E$ is likely to be similar across agonists (Lape

*et al.* 2008), the simplest change that could account for our observations would be a decrease in $E$. Extensive work (Chen *et al.* 2004; Chen & Huang, 2007) has investigated which residues mediate the effect of protons on GlyR, by mutating amino acids whose side chain may be protonated at the relevant pH range. These studies have identified a variety of His and Thr residues, scattered in all GlyR extracellular domains, from the binding site to the interface with the transmembrane domain, an area likely to be involved in signal transduction.

Is an impairment in gating the only change produced by low pH? By itself, a 3-fold decrease in efficacy is expected to reduce agonist potency only by about 1.5-fold (the cube root of the fold-change in efficacy; Colquhoun, 1998), which is between half and a third of the change we observed in whole-cell. However, this calculation assumes not only that three agonist molecules produce maximum opening (likely to be true; Beato *et al.* 2004; Burzomato *et al.* 2004), but also that changes in $EC_{50}$ values measured in whole-cell reflect purely changes in GlyR activation, a condition more difficult to satisfy. Nevertheless, we cannot exclude the possibility that acidic pH affects agonist binding to GlyR.

### Extracellular acidification impairs glycinergic synaptic signals

During synaptic transmission, receptors in the cleft are exposed for a brief time to a saturating neurotransmitter concentration (Clements, 1996). At glycinergic synapses onto mammalian spinal motoneurons, glycine reaches a peak concentration of 2.2–3.5 mM, which decays with a time constant of 0.6–0.9 ms (Beato, 2008). These conditions can be reproduced with fast agonist applications if recording from outside-out patches, where the limits to agonist diffusion posed by the unstirred layer are minimised. With this technique, we have showed that the decay kinetics of recombinant GlyR currents do reproduce those observed in spinal cord slices, provided intracellular chloride is kept low, to reproduce native conditions (Pitt *et al.* 2008).

Our current work shows that these glycinergic responses to concentration jumps were reduced by acidic extracellular pH, as there was both a reduction in the peak amplitude and a speeding up of deactivation, i.e. the decay of the current after the end of the 2-ms agonist pulse. For glycine, the peak amplitude was reduced by about 20% and the decay time constant speeded up by 2.7-fold (Table 4). These effects were similar to those observed for glycinergic mIPSC in dissociated spinal neurones from P14 rats by Li *et al.* (2003), who used a (symmetrical) high chloride medium in the patch pipette, probably to amplify the signal. This deviation from physiological conditions is unlikely to have made a

difference, because we found the effects of acidic pH to be reversible and approximately the same in magnitude in high and in low (10 mM) intracellular chloride. With low chloride, glycine peak was reduced by 40% and the decay time constant decreased by 2.4-fold (Table 4). Thus, the confounding effect of possible depletion of the chloride gradient across the patch are likely to be small and the effects of pH are likely to be present in native neurones with physiological (e.g. low) intracellular chloride. Similar effects were observed when we examined the behaviour of the heteromeric GlyR, that is likely to be the synaptic receptor in adult mammalian CNS. The peak of responses to glycine pulses was decreased to $73 \pm 8\%$ of control and their decay became 1.7-times faster (at 10 mM intracellular chloride). In both receptors and at all chloride concentrations we also detected a slower current onset in acidic pH. We cannot quantify the magnitude of this effect for glycine responses, given that in control conditions the rate of rise of the current is limited by the time course of the agonist exchange, rather than by the receptor. For the partial agonists we tested, the rise time was measurable in control conditions and slowed by approximately 50% in acidic pH. Finally, single-channel conductance was unchanged in acidic pH. Thus, the reduction in GlyR currents is due to changes in the kinetics of the GlyR, changes that, as we have seen, reduce the efficacy of agonists in equilibrium responses.

As we have seen, acidic pH has multiple effects on the responses to glycine concentration jumps and in order to have an overall view, it is probably best to consider what happens to the area under the curve of the current responses, as a proxy of charge transfer at the synapse. For the homomeric receptor, the area under the curve of the currents was reduced to $35 \pm 29\%$ of the original value ($n = 7$, 30 mM intracellular chloride) and recovered to $92 \pm 22\%$. This effect was maintained in experiments more relevant for synaptic conditions and with rat $\alpha 1\beta$ GlyRs the decrease at pH 6.4 was to $44 \pm 11\%$ of control (10 mM chloride, $n = 7$). The overall loss in the strength of glycinergic inhibition is thus likely to be substantial.

Other synaptic channels are also affected by acidic pH. For instance, glutamate channels are inhibited by acidic pH and the slower NMDA-mediated signals are more sensitive than AMPA currents (Traynelis & Cull-Candy, 1990). Inhibitory mechanisms mediated by GABA receptors are also affected by acidic pH, but whether their synaptic signals are depressed or enhanced depends on the exact composition of the channel, which is not precisely known for neurones in the caudal CNS, where glycinergic IPSC are most important. This makes it difficult to predict how the balance between fast and slow signals and between excitation and inhibition will change overall in acidic pH.

Our data show clearly that an acidic shift in extracellular pH comparable to that occurring in ischaemia or seizures causes a marked impairment in glycinergic responses, an impairment that involves a decrease in channel gating and weakens the impact of glycinergic synaptic signalling. This work poses the foundation for further investigation of the detailed mechanism of these effects.

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

## Additional information

### Data availability statement

The data that support the findings of this study are available from the corresponding author upon reasonable request.

### Competing interests

The authors declare that they have no conflict of interests.

### Author contributions

All authors conceived and designed the experiments; J.I. performed and analysed the experiments; J.I. and R.L. performed modelling calculations; J.I. and L.G.S. wrote the manuscript. All authors approved the final version of the manuscript and are accountable for all aspects of the work. All persons designated as authors qualify for authorship and all those who qualify for authorship are listed.

### Funding

Work was supported by an MRC project grant (MR/R009074/1). Project grant to L. Sivilotti.

### Acknowledgements

The authors are grateful to Dr Cali Hyde, Deltadot Bioanalysis Ltd, London NW1 0NH for carrying out the HPLC analysis of the agonists and to Dr George Papageorgiou, The Francis Crick Institute, 1 Midland Road, London NW1 1AT for carrying the purification of the taurine and GABA.

### Keywords

glycine receptors, pH changes, single channel

### Supporting information

Additional supporting information can be found online in the Supporting Information section at the end of the HTML view of the article. Supporting information files available:

**Peer Review History**
**Statistical Summary Document**

