## [Peer Review File · The Journal of physiology]

Acidic pH reduces agonist efficacy and responses to synaptic-like glycine applications in zebrafish $\alpha 1$ and rat $\alpha 1\beta$ recombinant glycine receptors

Josip Ivica, Remigijus Lape, and Lucia G Sivilotti

DOI: 10.1113/JP282171

Corresponding author(s): Lucia Sivilotti (l.sivilotti@ucl.ac.uk)

The following individual(s) involved in review of this submission have agreed to reveal their identity: Karin R. Aubrey (Referee #1); Yong Lu (Referee #2)

Review Timeline:

Submission Date:	21-Jun-2021
Editorial Decision:	19-Jul-2021
Resubmission Received:	19-Oct-2021
Editorial Decision:	04-Nov-2021
Revision Received:	12-Nov-2021
Accepted:	16-Nov-2021

Senior Editor: David Wyllie

Reviewing Editor: Jian Shi

Transaction Report:

Dear Professor Sivilotti,

Re: JP-RP-2021-281958 "Proton modulation of zebrafish recombinant $\alpha 1$ glycine receptors: decreased agonist efficacy and predicted reduction of glycinergic synaptic currents" by Josip Ivica, Remigijus Lape, and Lucia G Sivilotti

Thank you for submitting your manuscript to The Journal of Physiology. It has been assessed by a Reviewing Editor and by 2 Referees and the reports are copied below.

Please let your co-authors know of the following editorial decision as quickly as possible.

As you will see, in its current form, the manuscript is not acceptable for publication in The Journal of Physiology. In comments to me, the Reviewing Editor expressed interest in the potential of this study, but much work still needs to be done (and this may include new experiments) in order to satisfactorily address the concerns raised in the reports.

In view of this interest, I would like to offer you the opportunity to carry out all of the changes requested in full, and to resubmit a new manuscript using the "Submit Special Case Resubmission for JP-RP-2021-281958..." on your homepage.

We cannot, of course, guarantee ultimate acceptance at this stage as the revisions required are substantial. However, we encourage you to consider the requested changes and resubmit your work to us if you are able to complete or address all changes.

A new manuscript would be renumbered and redated, but the original referees would be consulted wherever possible. An additional referee's opinion could be sought, if the Reviewing Editor felt it necessary. A full response to each of the reports should be uploaded with a new version.

I hope that the points raised in the reports will be helpful to you.

Yours sincerely,

David Wyllie
Senior Editor
The Journal of Physiology

EDITOR COMMENTS

Reviewing Editor:

In this manuscript the authors investigated the mechanisms by which glycine receptor currents are modulated by pH6.4 through single-channel and whole-cell electrophysiological recordings in overexpression system. The authors demonstrated that acidification reduced channel open probability of GlyRs, and reduced the channel responses. Though referees appreciate that it is an interesting study with nice electrophysiological recordings, some concerns remain.

Specific comments:

One major concern is how relevant such study on zebrafish glycine receptor is to physiological/pathophysiological functions and what novel physiological insight this study could generate. In addition, the chloride concentration used for electrophysiological recordings was far higher than the physiological one, which potentially casts doubt on whether the study is relevant to physiology or pathophysiology.

Other specific comments include clarification of experimental details, data analysis and typos etc. Please see both referee reports for details.

Senior Editor:

Your manuscript has been assessed by two expert reviewers and a Reviewing Editor. Each consider the work presented as of a very high standard but also raise concerns about its wider applicability and to our understanding of physiology, given the nature of the glycine receptors studied and the fact that the work is carried out in an expression system. In addition the "low" chloride internal solution you use is 30 mM - this concentration of intracellular Cl⁻ is only seen in very early development in mammalian neurons and rapidly falls to 8 - 10 mM in mature neurons. Thus, in a revised submission it is essential that you ensure that you address the major concerns of the reviewers. If you decide to submit a revised version, it will be considered a new submission but we will endeavour, but cannot guarantee, to have the same reviewers assess your work. It is therefore possible that new points will be raised during any subsequent review process.

REFeree COMMENTS

Referee #1:

The paper by Ivica, Lape and Sivilotti look in detail at the mechanisms by which glycine receptor currents are modulated by pH6.4 vs physiological pH and pH8. They express homomeric zebrafish glycine receptors in HEK cells and make electrophysiological recordings of single-channel and outside-out patch glycine currents. They compare the effect of acidic pH on currents stimulated by glycine, as well as partial agonists taurine and beta-alanine. They demonstrate that GlyR open probability is reduced in acidic pH, with no change in conductance. Then, in recordings of outside-out patch responses to fast agonist applications the authors measured a reduction in peak current and change in current kinetic in acidic pH, and rule out the possibility that changes to intracellular chloride contribute to this effect. The modulation of GlyR by acidic pH has been previously reported. The advance in the field is the demonstration this effect occurs as a result of an alteration of GlyR open probability and the direct link to synaptic-like current reductions. In addition, the authors compare this with acidic pH effects on GlyR activation by 2 GlyR partial agonists and confirm that intracellular chloride is not involved. The study design and data are clean and well presented.

Major comment:

The authors need to explain why this sort of work is useful and relevant. How might it be used to grow the research area? In particular, justify why was the study done using zebrafish glycine receptors and alpha1 homomers? Given mammalian brain glycine receptors are most commonly made from a1b subunits, is this the most relevant approach? The work could be extended by including heteromeric receptors.

Minor comments:

There are a few typographical errors in the abstract.

Introduction:

What is the significance of ASIC being activated? and indicate what "important details" are currently lacking.

Methods:

Do you have evidence that the U-tube application of glycine is consistent wrt to time on and off? This is important for the interpretation of the current kinetic changes measured.

Results and Discussion:

GABA is mentioned as a partial agonist of GlyR. This is not the case for mammalian glycine receptors and should be indicated and referenced. It also highlights that the zebrafish glycine receptor has different agonist binding to mammalian GlyR. In addition, in the discussion you mention fish GlyR is "easier to open". Are there other functional differences between species too? Do your results translate to the mammalian receptor? This should be highlighted and discussed.

Referee #2:

General Comments:

The manuscript by Ivica et al. has convincingly demonstrated that acidification reduced channel open probability of GlyRs, and reduced the channel responses, in cultured cells. The study therefore has the implication that acidic pH, which occurs under physiological conditions in the CNS (and even enhanced in pathological conditions), regulates the synaptic inhibitory strength mediated by glycine. The experiments are well designed and executed. Recording quality is high. Data analyses and presentations are thorough and clear. The manuscript is well written and easy to follow. Overall, the study is another continuous excellent effort on the research theme directed by the senior author. Clarifications of a few major points and a number of minor issues will strengthen the manuscript.

Major Points:

The first issue is regarding the intracellular Cl⁻ concentration. When equal Cl⁻ concentration (131.1 mM) is used for both internal and external solutions, the situation is absolutely artificial (not physiological), which has been stated as such in the manuscript. However, the authors used an intracellular Cl⁻ concentration of 30 mM, and claimed that this is low concentration. While 30 mM is substantially lower than 131.1 mM, the readers should not be misled to assume that this is physiological. Relatively high Cl⁻ concentration (about 30 mM) exists primarily in developing neurons. In mature CNS neurons, most of them have much lower intracellular Cl⁻ concentration (around 10 mM), giving rise to membrane hyperpolarization when Cl⁻ channels such as GlyR and GABAR are activated, due to influx of Cl⁻ ions. The authors need to make this point clearer in the context of using 30 mM Cl⁻ in their "low Cl⁻" recordings.

The second issue is on the Discussion. A large part of the Discussion is kind of reiterating the results. The readers may be

left unsatisfied in terms of understanding the mechanisms and physiological implications. It is understood that the mechanisms underlying the reduction of GlyR or GABAR responses due to acidification may have been presented in previous works. It will be helpful to re-state them and cite the papers while discussing the results from the current study. Similarly, in terms of physiological implications, more extensive discussion, even if they are speculations, will stimulate interest and thought process for readers, making the current study more impactful.

Specific Points:

1. Page 4, the second half of the last sentence "clearly show..." is not clearly written, likely due to grammatical errors.
2. Page 7, 3rd line from bottom: change "by 10 s" to "every 10 s".
3. Page 8, line 6-7: how is the weighted time constant calculated?
4. Fig 1: at pH 7.4, what is the response to 30 mM Glycine application? Same as that to 10 mM Glycine (saturated)?
5. Fig 2 legend: what is "eg" in the last phrase?
6. Fig 3C: what does the dashed line indicate?
- c7. Fig 4 labeling and Table 2: to be precise and consistent, label and report the exact Cl⁻ concentration (131.1 mM), instead of 131 mM.
8. Decay time constant and Deactivation constant are used in different figures and/or tables. Pick and use one consistently.
9. Page 23, line 12, add a period after 0.93.

END OF COMMENTS

We are grateful for the supportive comments of the Editor and the referees, cf. the interest of our work and the quality of our electrophysiology recordings.

We agree that the original manuscript was missing data of greater relevance to synapses, such as recordings from heteromeric receptors and at physiologically low (10 mM intracellular chloride) and because of that we have carried out a substantial number of new experiments in these conditions, where we found that the effect of pH was maintained. We feel the paper is stronger for this changes.

EDITOR COMMENTS Reviewing Editor:

In this manuscript the authors investigated the mechanisms by which glycine receptor currents are modulated by pH6.4 through single-channel and whole-cell electrophysiological recordings in overexpression system. The authors demonstrated that acidification reduced channel open probability of GlyRs, and reduced the channel responses. Though referees appreciate that it is an interesting study with nice electrophysiological recordings, some concerns remain.

Specific comments:

*One major concern is how relevant such study on **zebrafish** glycine receptor is to physiological/pathophysiological functions and what novel physiological insight this study could generate. In addition, **the chloride** concentration used for electrophysiological recordings was far higher than the physiological one, which potentially casts doubt on whether the study is relevant to physiology or pathophysiology.*

Other specific comments include clarification of experimental details, data analysis and typos etc. Please see both referee reports for details.

Senior Editor:

Your manuscript has been assessed by two expert reviewers and a Reviewing Editor. Each consider the work presented as of a very high standard but also raise concerns about its wider applicability and to our understanding of physiology, given the nature of the glycine receptors studied and the fact that the work is carried out in an expression system. In addition the "low" chloride internal solution you use is 30 mM - this concentration of intracellular Cl⁻ is only seen in very early development in mammalian neurons and rapidly falls to 8 - 10 mM in mature neurons. Thus, in a revised submission it is essential that you ensure that you address the major concerns of the reviewers. If you decide to submit a revised version, it will be considered a new submission but we will endeavour, but cannot guarantee, to have the same reviewers assess your work. It is therefore possible that new points will be raised during any subsequent review process.

REFeree COMMENTS

Referee #1:

The paper by Ivica, Lape and Sivilotti look in detail at the mechanisms by which glycine receptor currents are modulated by pH6.4 vs physiological pH and pH8. They express homomeric zebrafish glycine receptors in HEK cells and make electrophysiological recordings of single-channel and outside-out patch glycine currents. They compare the effect of acidic pH on currents stimulated by glycine, as well as partial agonists taurine and beta-alanine. They demonstrate that GlyR open probability is reduced in acidic pH, with no change in conductance. Then, in recordings of outside-out patch responses to fast agonist applications the authors measured a reduction in peak current and change in current kinetic in acidic pH, and rule out the possibility that changes to intracellular chloride contribute to this effect. The modulation of GlyR by acidic pH has been previously reported. The advance in the field is the demonstration this effect occurs as a result of an alteration of GlyR open probability and the direct link to synaptic-like current reductions. In addition, the authors compare this with acidic pH effects on GlyR activation by 2 GlyR partial agonists and confirm that intracellular chloride is not involved. The study design and data are clean and well presented.

Major comment:

The authors need to explain why this sort of work is useful and relevant. How might it be used to grow the research area? In particular, justify why was the study done using zebrafish glycine receptors and alpha1 homomers? Given mammalian brain glycine receptors are most commonly made from alpha1b subunits, is this the most relevant approach? The work could be extended by including heteromeric receptors.

This has all been done. Our new experiments confirm that modest acidic changes in extracellular pH inhibit mammalian synaptic GlyRs in conditions that are relevant to synaptic transmission (concentration jumps and low intracellular chloride – 10 mM). These results are shown in a new figure (Figure 7) and in Table 4.

Minor comments:

There are a few typographical errors in the abstract.

Introduction:

What is the significance of ASIC being activated?

We have removed this sentence. As the referee points out, it is not relevant in our heterologous expression system.

and indicate what "important details" are currently lacking.

There is only one study (Li et al., 2003) that examines the effect of pH changes on synaptic GlyR currents, and this is in acutely dissociated spinal neurones, where the glycinergic mIPSC is produced by boutons, but the receptor effects are not examined. We have toned down this, and the details that the picture lacks are listed in the following sentences. Changes in agonist efficacy on the GlyR were never assessed in the existing literature.

Methods:

Do you have evidence that the U-tube application of glycine is consistent wrt to time on and off? This is important for the interpretation of the current kinetic changes measured.

The kinetic measurements are not done on the U-tube whole cell recordings, but on the piezo stepper applications to outside out-patches. We have inserted a sentence in the Methods to state this, as it was obviously unclear.

Results and Discussion:

GABA is mentioned as a partial agonist of GlyR. This is not the case for mammalian glycine receptors and should be indicated and referenced.

It also highlights that the zebrafish glycine receptor has different agonist binding to mammalian GlyR. In addition, in the discussion you mention fish GlyR is "easier to open". Are there other functional differences between species too? Do your results translate to the mammalian receptor? This should be highlighted and discussed.

We have looked in some detail at the agonist potency and efficacy on GlyRs from different species, as structural work by our collaborators was done in the zebrafish isoform. We have recently shown that zebrafish and human alpha1 GlyR have similar agonist sensitivity (Ivica et al., J. Biol. Chem. 296 100387, 2021). This finding is not surprising given that two receptors share 94% of amino acid similarity. Most of the differences are found in large intracellular domain and in transmembrane domain 4, and none of these are in the binding loops. GABA is a very weak agonist, the weakest of the ones examined in our paper (glycine, beta-alanine and taurine) and elicits a maximum Popen of 9 +/- 8% in the human alpha1 GlyR.

With these small responses to GABA, it is no surprise that the earliest descriptions of GlyR pharmacology failed to observe an agonist effect of GABA. However, a partial agonist action was reported for GABA by De Saint Jan et al, in JPhysiol 2001 (human GlyR expressed in oocytes) and Jonas et al., 1998 (rat spinal motoneurons). Note also that GABA is co-released with glycine in rat auditory synapses and speeds up the decay of the glycine IPSCs to become faster than glycine deactivation (Lu and Trussell, 2008). For this effect to occur, clearly GABA must bind to the GlyR and act as a very weak agonist or antagonist, and our data show that it is the former.

We use a panel of agonists because the partial agonists are more sensitive than the very efficacious agonist glycine to changes in efficacy and in activation speed (and state this on page 14).

We have now extended our work to mammalian GlyR, and confirmed our findings, making a discussion of species differences irrelevant. We recently examined the agonist pharmacology

of zebrafish and human alpha1 GlyR in detail and found that agonists have the same order of potency and efficacy in these two receptors. The main differences were in the magnitude of the maximum response seen and this could be manipulated by engineering swaps of the intracellular domain, arguing for a species difference in the basal allosteric constant and the propensity to gate, but not in the agonist specificity in the binding site.

We had examined these differences between zebrafish and mammalian channel in our recent JBC paper and are reluctant to repeat here the discussion.

Referee #2:

General Comments:

The manuscript by Ivica et al. has convincingly demonstrated that acidification reduced channel open probability of GlyRs, and reduced the channel responses, in cultured cells. The study therefore has the implication that acidic pH, which occurs under physiological conditions in the CNS (and even enhanced in pathological conditions), regulates the synaptic inhibitory strength mediated by glycine. The experiments are well designed and executed. Recording quality is high. Data analyses and presentations are thorough and clear. The manuscript is well written and easy to follow. Overall, the study is another continuous excellent effort on the research theme directed by the senior author. Clarifications of a few major points and a number of minor issues will strengthen the manuscript.

Major Points:

The first issue is regarding the intracellular Cl⁻ concentration. When equal Cl⁻ concentration (131.1 mM) is used for both internal and external solutions, the situation is absolutely artificial (not physiological), which has been stated as such in the manuscript. However, the authors used an intracellular Cl⁻ concentration of 30 mM, and claimed that this is low concentration. While 30 mM is substantially lower than 131.1 mM, the readers should not be misled to assume that this is physiological. Relatively high Cl⁻ concentration (about 30 mM) exists primarily in developing neurons. In mature CNS neurons, most of them have much lower intracellular Cl⁻ concentration (around 10 mM), giving rise to membrane hyperpolarization when Cl⁻ channels such as GlyR and GABAR are activated, due to influx of Cl⁻ ions. The authors need to make this point clearer in the context of using 30 mM Cl⁻ in their "low Cl⁻" recordings.

This has all been done- we have extended the original experiments in zebrafish channels to 10 mM intracellular chloride and carried out new experiments with rat heteromeric receptors at a range of intracellular chloride concentrations (131.1, 30 and 10 mM). The effect is maintained in these conditions. It is still useful to have seen similar effects of pH on concentration jumps with higher chloride concentrations, as it reassures us that a contribution by chloride gradient shifts is unlikely to be important. All the Figures have been clearly labelled, with the actual chloride concentration, as the referee suggested

The second issue is on the Discussion. A large part of the Discussion is kind of reiterating the results. The readers may be left unsatisfied in terms of understanding the mechanisms and

physiological implications. It is understood that the mechanisms underlying the reduction of GlyR or GABAR responses due to acidification may have been presented in previous works. It will be helpful to re-state them and cite the papers while discussing the results from the current study. Similarly, in terms of physiological implications, more extensive discussion, even if they are speculations, will stimulate interest and thought process for readers, making the current study more impactful.

We have added a paragraph to the Discussion where we examine the effects of acidic pH on the balance between excitation and inhibition and on fast vs. slower signals.

Specific Points:

Thank you for spotting these! The following points have all been changed as you suggest.

1. Page 4, the second half of the last sentence "clearly show..." is not clearly written, likely due to grammatical errors.

2. Page 7, 3rd line from bottom: change "by 10 s" to "every 10 s".

3. Page 8, line 6-7: how is the weighted time constant calculated?

4. Fig 1: at pH 7.4, what is the response to 30 mM Glycine application? Same as that to 10 mM Glycine (saturated)?

Yes – we used 30 mM at lower pH, just in case the negative modulation made 10 mM submaximal). We say this in the legend.

5. Fig 2 legend: what is "eg" in the last phrase?

6. Fig 3C: what does the dashed line indicate? It is the closed level, now added

7. Fig 4 labeling and Table 2: to be precise and consistent, label and report the exact Cl⁻ concentration (131.1 mM), instead of 131 mM.

8. Decay time constant and Deactivation constant are used in different figures and/or tables. Pick and use one consistently.

9. Page 23, line 12, add a period after 0.93.

Dear Professor Sivilotti,

Re: JP-RP-2021-282171X "Acidic pH reduces agonist efficacy and responses to synaptic-like glycine applications in zebrafish $\alpha 1$ and rat $\alpha 1\beta$ recombinant glycine receptors" by Josip Ivica, Remigijus Lape, and Lucia G Sivilotti

Thank you for submitting your manuscript to The Journal of Physiology. It has been assessed by a Reviewing Editor and by 2 expert Referees and I am pleased to tell you that it is considered to be acceptable for publication following satisfactory revision.

The reports are copied at the end of this email. Please address all of the points and incorporate all requested revisions, or explain in your Response to Referees why a change has not been made.

NEW POLICY: In order to improve the transparency of its peer review process The Journal of Physiology publishes online as supporting information the peer review history of all articles accepted for publication. Readers will have access to decision letters, including all Editors' comments and referee reports, for each version of the manuscript and any author responses to peer review comments. Referees can decide whether or not they wish to be named on the peer review history document.

Authors are asked to use The Journal's premium BioRender (<https://biorender.com/>) account to create/redrawn their Abstract Figures. Information on how to access The Journal's premium BioRender account is here: <https://physoc.onlinelibrary.wiley.com/journal/14697793/biorender-access> and authors are expected to use this service. This will enable Authors to download high-resolution versions of their figures.

I hope you will find the comments helpful and have no difficulty returning your revisions within 4 weeks.

Your revised manuscript should be submitted online using the links in Author Tasks Link Not Available.

Any image files uploaded with the previous version are retained on the system. Please ensure you replace or remove all files that have been revised.

REVISION CHECKLIST:

- Article file, including any tables and figure legends, must be in an editable format (eg Word)
- Abstract figure file (see above)
- Statistical Summary Document
- Upload each figure as a separate high quality file
- Upload a full Response to Referees, including a response to any Senior and Reviewing Editor Comments;
- Upload a copy of the manuscript with the changes highlighted.

- A potential 'Cover Art' file for consideration as the Issue's cover image;
- Appropriate Supporting Information (Video, audio or data set https://jp.msubmit.net/cgi-bin/main.plex?form_type=display_requirements#supp).

To create your 'Response to Referees' copy all the reports, including any comments from the Senior and Reviewing Editors, into a Word, or similar, file and respond to each point in colour or CAPITALS and upload this when you submit your revision.

I look forward to receiving your revised submission.

If you have any queries please reply to this email and staff will be happy to assist.

Yours sincerely,

David Wyllie

EDITOR COMMENTS

Reviewing Editor:

This revised manuscript investigated the mechanisms underlying how pH modulates glycine receptors. It is discovered that the decrease of acidic pH by 1 unit would reduce glycine receptor open probability and then limit glycine signalling. Under physiological chloride gradients, these effects could take place not only in zebrafish glycine receptor, but also in mammalian heteromeric glycine receptor.

The revision has responded well to the previous comments, which has been agreed by the referees. But there are still some minor comments the authors need to address. These minor comments include the clarification of experimental details, data interpretation and statistical significance, and typo corrections.

Senior Editor:

Comments for Authors to ensure the paper complies with the Statistics Policy:

While significance is detailed correctly with exact p values etc in the text, the details are not provided in figures/figure legends. Some panels are presented without any indication that there are statistical differences and one needs to read the main text to find this out. Can these please be indicated in the figures and details (p values) included in Fig legends (where appropriate)

Comments to the Author:

Thank-you for the revision of your manuscript. As you will see a few comments have been raised by the referees which I think can be easily addressed. I have also noted there needs to be further clarity in some of the data presentation. If these points can be addressed and a further revised manuscript submitted I will then proceed to final acceptance.

REFEREE COMMENTS

Referee #1:

The authors have investigated the mechanisms by which pH modulates glycine receptors. They carry out careful experiments which show that acidic dips in pH by 1 unit limit glycine signalling by reducing glycine channel open probability. They go on to show that this phenomenon is likely to occur under physiological chloride gradients and at a common form of mammalian glycine receptors. The authors have responded comprehensively to previous comments.

Minor correction and questions:

- Is it strange that the data at pH8.4 is so variable in this system? Why do you think this is so? what do you think is happening?

- Significance should be indicated in figures 4BCD-5BC - 6CD and 7CD

Minor typographical corrections:

- I suggest removing (parenthesis) from the abstract

- Check for inverted phases eg. characterised also = also characterised/ acidification changed also the time = Acidification also changed.

- finish sentence/typo page 20 "experiments at , 30mM intracellular chloride (a compromise condition where responses are still large) and tested the effects of pH changes on glycine and taurine."

- page 23, move "(table 3)" to the end of the sentence

Referee #2:

The manuscript has been revised well in response to my reviews. However, there exists one more minor concern. In the newly generated Fig 7, when 10 mM intracellular Cl⁻ was used, what was the holding potential for the voltage clamp experiment? If the same holding potential (-40 mV) was used, as stated in the Methods, one would expect to observe outward GlyR current, because the equilibrium potential for Cl⁻ now (about -64 mV) is more negative than the holding potential. Therefore, I assume the holding potential was changed to a level more negative than -64 mV. If so, this needs to be stated clearly when Fig 7 is described.

We are grateful for the quick and supportive reply by the editors and referees- we have carried out all the changes requested.

Editor Comments

Reviewing Editor:

This revised manuscript investigated the mechanisms underlying how pH modulates glycine receptors. It is discovered that the decrease of acidic pH by 1 unit would reduce glycine receptor open probability and then limit glycine signalling. Under physiological chloride gradients, these effects could take place not only in zebrafish glycine receptor, but also in mammalian heteromeric glycine receptor.

The revision has responded well to the previous comments, which has been agreed by the referees. But there are still some minor comments the authors need to address. These minor comments include the clarification of experimental details, data interpretation and statistical significance, and typo corrections.

Senior Editor:

Comments for Authors to ensure the paper complies with the Statistics Policy:

While significance is detailed correctly with exact p values etc in the text, the details are not provided in figures/figure legends. Some panels are presented without any indication that there are statistical differences and one needs to read the main text to find this out. Can these please be indicated in the figures and details (p values) included in Fig legends (where appropriate)

- We now indicate significance in the figures and refer in the legend to text or tables for p values.

Comments to the Author:

Thank-you for the revision of your manuscript. As you will see a few comments have been raised by the referees which I think can be easily addressed. I have also noted there needs to be further clarity in some of the data presentation. If these points can be addressed and a further revised manuscript submitted I will then proceed to final acceptance.

REFEREE COMMENTS

Referee #1:

The authors have investigated the mechanisms by which pH modulates glycine receptors. They carry out careful experiments which show that acidic dips in pH by 1 unit limit glycine signalling by reducing glycine channel open probability. They go on to show that this phenomenon is likely to occur under physiological chloride gradients and at a common form of mammalian glycine receptors. The authors have responded comprehensively to previous comments.

Minor correction and questions:

- Is it strange that the data at pH8.4 is so variable in this system? Why do you think this is so? what do you think is happening?

- The short answer is that we don't know. If pushed, we could speculate that, while we are using SD as a measure of scatter independent of n, the smaller number of experiments happened by chance to be quite dispersed at pH 8.4. However, what matters is that clearly

the effect of going one unit more basic was small, which is why we chose to concentrate on the more consistent and pathophysiologically interesting acidic pH.

- Significance should be indicated in figures 4BCD-5BC - 6CD and 7CD

- Done

Minor typographical corrections:

- I suggest removing (parenthesis) from the abstract

- Done

- Check for inverted phases eg. characterised also = also characterised/ acidification changed also the time = Acidification also changed.

- Done

- finish sentence/typo page 20 "experiments at , 30mM intracellular chloride (a compromise condition where responses are still large) and tested the effects of pH changes on glycine and taurine."

- Done

- page 23, move "(table 3)" to the end of the sentence

- Done

Referee #2:

The manuscript has been revised well in response to my reviews. However, there exists one more minor concern. In the newly generated Fig 7, when 10 mM intracellular Cl⁻ was used, what was the holding potential for the voltage clamp experiment? If the same holding potential (-40 mV) was used, as stated in the Methods, one would expect to observe outward GlyR current, because the equilibrium potential for Cl⁻ now (about -64 mV) is more negative than the holding potential. Therefore, I assume the holding potential was changed to a level more negative than -64 mV. If so, this needs to be stated clearly when Fig 7 is described.

- The holding at -40 mV refers only to the whole cell experiments. All concentration jumps were done at a nominal holding of -100 mV (-102, -110 and -113 mV when corrected for junction potential in experiments with 131.1, 30 and 10 mM internal chloride). This was stated only in the methods and is now mentioned for clarity in the legends of the appropriate figures and tables.

Dear Dr Sivilotti,

Re: JP-RP-2021-282171XR1 "Acidic pH reduces agonist efficacy and responses to synaptic-like glycine applications in zebrafish $\alpha 1$ and rat $\alpha 1\beta$ recombinant glycine receptors" by Josip Ivica, Remigijus Lape, and Lucia G Sivilotti

I am pleased to tell you that your paper has been accepted for publication in The Journal of Physiology.

NEW POLICY: In order to improve the transparency of its peer review process The Journal of Physiology publishes online as supporting information the peer review history of all articles accepted for publication. Readers will have access to decision letters, including all Editors' comments and referee reports, for each version of the manuscript and any author responses to peer review comments. Referees can decide whether or not they wish to be named on the peer review history document.

Are you on Twitter? Once your paper is online, why not share your achievement with your followers. Please tag The Journal (@jphysiol) in any tweets and we will share your accepted paper with our 23,000+ followers!

The last Word version of the paper submitted will be used by the Production Editors to prepare your proof. When this is ready you will receive an email containing a link to Wiley's Online Proofing System. The proof should be checked and corrected as quickly as possible.

Authors should note that it is too late at this point to offer corrections prior to proofing. The accepted version will be published online, ahead of the copy edited and typeset version being made available. Major corrections at proof stage, such as changes to figures, will be referred to the Reviewing Editor for approval before they can be incorporated. Only minor changes, such as to style and consistency, should be made a proof stage. Changes that need to be made after proof stage will usually require a formal correction notice.

All queries at proof stage should be sent to TJP@wiley.com

Yours sincerely,

David Wyllie
Senior Editor
The Journal of Physiology

P.S. - You can help your research get the attention it deserves! Check out Wiley's free Promotion Guide for best-practice recommendations for promoting your work at www.wileyauthors.com/eoo/guide. And learn more about Wiley Editing Services which offers professional video, design, and writing services to create shareable video abstracts, infographics, conference posters, lay summaries, and research news stories for your research at www.wileyauthors.com/eoo/promotion.

*** IMPORTANT NOTICE ABOUT OPEN ACCESS ***

Information about Open Access policies can be found here <https://physoc.onlinelibrary.wiley.com/hub/access-policies>

To assist authors whose funding agencies mandate public access to published research findings sooner than 12 months after publication The Journal of Physiology allows authors to pay an open access (OA) fee to have their papers made freely available immediately on publication.

You will receive an email from Wiley with details on how to register or log-in to Wiley Authors Services where you will be able to place an OnlineOpen order.

You can check if your funder or institution has a Wiley Open Access Account here <https://authorservices.wiley.com/author-resources/Journal-Authors/licensing-and-open-access/open-access/author-compliance-tool.html>

Your article will be made Open Access upon publication, or as soon as payment is received.

If you wish to put your paper on an OA website such as PMC or UKPMC or your institutional repository within 12 months of publication you must pay the open access fee, which covers the cost of publication.

OnlineOpen articles are deposited in PubMed Central (PMC) and PMC mirror sites. Authors of OnlineOpen articles are permitted to post the final, published PDF of their article on a website, institutional repository, or other free public server,

immediately on publication.

Note to NIH-funded authors: The Journal of Physiology is published on PMC 12 months after publication, NIH-funded authors DO NOT NEED to pay to publish and DO NOT NEED to post their accepted papers on PMC.

EDITOR COMMENTS

Reviewing Editor:

Ethics Concerns:

No concern as there was no any animal or human tissue used for the study.

Comments to the Author:

Thank you for the revision. All concerns or questions raised by the referees and editors have been addressed.

Senior Editor:

Comments to the Author:

Many thanks for the further clarifications. Happy to accept your manuscript.

END OF COMMENTS

2nd Confidential Review

12-Nov-2021